# A nonstructural protein encoded by a rice reovirus induces an incomplete autophagy to promote viral spread in insect vectors

**Dongsheng Jia**[1☯]**, Qifu Liang**[1☯]**, Huan Liu**[1]**, Guangjun Li**[1]**, Xiaofeng Zhang**[1]**, Qian Chen**[1]**, Aiming Wang**[2]**, Taiyun Wei**[1]*

**1** Vector-borne Virus Research Center, State Key Laboratory of Ecological Pest Control for Fujian and Taiwan Crops, Fujian Agriculture and Forestry University, Fuzhou, Fujian, China, **2** London Research and Development Centre, Agriculture and Agri-Food Canada, London, Ontario, Canada

☯ These authors contributed equally to this work.

* weitaiyun@fafu.edu.cn

## Abstract

Viruses can hijack autophagosomes as the nonlytic release vehicles in cultured host cells. However, how autophagosome-mediated viral spread occurs in infected host tissues or organs *in vivo* remains poorly understood. Here, we report that an important rice reovirus, rice gall dwarf virus (RGDV) hijacks autophagosomes to traverse multiple insect membrane barriers in the midgut and salivary gland of leafhopper vector to enhance viral spread. Such virus-containing double-membraned autophagosomes are prevented from degradation, resulting in increased viral propagation. Mechanistically, viral nonstructural protein Pns11 induces autophagy and embeds itself in the autophagosome membranes. The autophagy-related protein 5 (ATG5)-ATG12 conjugation is essential for initial autophagosome membrane biogenesis. RGDV Pns11 specifically interacts with ATG5, both *in vitro* and *in vivo*. Silencing of ATG5 or Pns11 expression suppresses ATG8 lipidation, autophagosome formation, and efficient viral propagation. Thus, Pns11 could directly recruit ATG5-ATG12 conjugation to induce the formation of autophagosomes, facilitating viral spread within the insect bodies. Furthermore, Pns11 potentially blocks autophagosome degradation by directly targeting and mediating the reduced expression of N-glycosylated Lamp1 on lysosomal membranes. Taken together, these results highlight how RGDV remodels autophagosomes to benefit viral propagation in its insect vector.

## Author summary

Numerous plant viruses replicate inside the cells of their insect vectors. Here, we demonstrate that the progeny virions of rice gall dwarf virus in leafhopper vector are engulfed within virus-induced double-membraned autophagosomes. Such autophagosomes are modified to evade degradation, thus can be persistently exploited by viruses to safely transport virions across multiple insect membrane barriers. Viral nonstructural protein Pns11 induces the formation of autophagosomes via interaction with ATG5, and

**Funding:** This project was supported by funds from the National Natural Science Foundation of China to TW under grant number 31920103014 (http://www.nsfc.gov.cn/), the National Natural Science Foundation of China to DJ under grant number 31970160 (http://www.nsfc.gov.cn/), the Natural Science Foundation of Fujian Province to DJ under grant number 2020J06015 (http://xmgl.kjt.fujian.gov.cn/loginSignout.do), the Science, Technology and Innovation Fundation of Fujian Agriculture and Forestry University to TW under grant number CXZX2019004K (https://kyy.fafu.edu.cn/), the Science, Technology and Innovation Fundation of Fujian Agriculture and Forestry University to DJ under grant number CXZX2019021G (https://kyy.fafu.edu.cn/). The funders had no role in study design, data collection and analysis, decision to publish, or preparation of the manuscript.

**Competing interests:** The authors have declared that no competing interests exist.

potentially blocks autophagosome degradation via mediating the reduced expression of N-glycosylated Lamp1 on lysosomal membranes. For the first time, we reveal that a non-structural protein encoded by a persistent plant virus can induce an incomplete autophagy to benefit viral propagation in its insect vectors.

## Introduction

Autophagy is a multistep, conserved process by which cytoplasmic components, such as damaged organelles and foreign pathogens, become enveloped into double-membraned autophagosomes and are shuttled to lysosomes for degradation in eukaryotic cells [1]. This process begins with the formation of membrane crescents, called phagophores, which then extend to form enclosed double-membraned autophagosomes [1]. Autophagy-related proteins (ATGs) and two ubiquitin-like conjugation systems mediate phagophore elongation and the subsequent generation of double-membraned autophagosomes. The first conjugation system entails ATG8-I, which is constitutively processed by ATG4 to expose conserved glycine residues to produce the phospholipid ATG8-II, and is conjugated to the membrane lipid phosphatidylethanolamine (PE) at the autophagosome membranes [2,3]. ATG8-PE is involved in the elongation and closure of the autophagosome membranes [1]. The second conjugation system contributes to the coupling of ATG12 with ATG5 for forming a covalently linked heterodimer on the phagophore membranes [3,4]. This conjugate is mobilized to the phagophore to produce E3-like activity toward the ATG8-PE complex [1]. Finally, the autophagosomes are fused with lysosomes to form autolysosomes, where the cytoplasmic materials or invading pathogens are degraded [1]. Lysosome-associated membrane protein (Lamp) consists of a polypeptide core of ~40 kDa and is decorated by a significant number (16–20) of N-glycans [5]. Lamp protects lysosomal membranes from lysosomal hydrolases that normally digest membrane components [6]. It is generally believed that host cells use the autophagy as a defense against invading viruses [7,8]. For example, autophagy has been proved to play a direct antiviral role against the arbovirus vesicular stomatitis virus in the model organism *Drosophila* [9].

Viruses have also evolved strategies to subvert autophagy by targeting autophagy machinery. For example, Herpes simplex virus type 1 neurovirulence protein prevents the phagophore formation by binding to Beclin 1, a key factor involved in the elongation of the phagophore membranes [10]. Human herpes virus-8 can suppress autophagy by preventing ATG3 from binding and processing ATG8 during autophagosome elongation [11]. Barley stripe mosaic virus (BSMV) γb protein subverts autophagy to promote viral infection by disrupting the interaction of ATG7 with ATG8 [12]. More importantly, viruses have also evolved strategies to inhibit the fusion of autophagosomes with lysosomes, thus directly exploiting autophagosomes as viral replication sites or as the nonlytic release vehicles in host cells [3,13,14]. Currently, most of such studies are limited to cultured host cells *in vitro*; how autophagosome-mediated viral spread occurs in infected host tissues or organs *in vivo* remains poorly understood.

Many viral pathogens of significant consequences to global health or agricultural problems are transmitted by insect vectors. Several arthropod-borne plant viruses are highly evolved and adapted for persistent infection and maintenance in their insect vectors. Such viruses circulate throughout the insect bodies and induce a variety of cellular responses such as autophagy or apoptosis that modulate the efficiency of viral transmission [15]. To date, only three reports on autophagy in plant viruses-infected insect cells have been published. In 2016, Wang et al. reported that the begomovirus tomato yellow leaf curl virus (TYLCV) could activate autophagy in whitefly vectors to resist excessive viral replication [16]. Similarly, a recently published study

demonstrated that the fijivirus rice black-streaked dwarf virus (RBSDV) infection could activate autophagy in planthopper vector, leading to inhibition of viral transmission [17]. In 2017, we reported that the phytoreovirus rice gall dwarf virus (RGDV) induced the formation of autophagosomes for carrying RGDV virions to facilitate viral infection in insect vector cells [18]. Thus, RGDV has evolved elaborate mechanisms to evade antiviral autophagy, or even to activate certain forms of selective autophagy for successful infection and replication within its insect vector.

RGDV, an important rice reovirus, causes epidemic outbreaks and extensive rice yield losses in Asian rice-growing countries, and is mainly transmitted by green rice leafhopper, *Recilia dorsalis*, in a persistent-propagative manner [19,20]. The 12 segments of RGDV genome encode 6 structural proteins (P1, P2, P3, P5, P6 and P8) and 6 nonstructural proteins (Pns4, Pns7, Pns9, Pns10, Pns11 and Pns12). Among these proteins, P8 is the major outer capsid protein, Pns7, Pns9 and Pns12 are the components of the viroplasm, Pns11 is a viral RNA-silencing suppressor, and the functions of Pns4 and Pns10 are unknown [21–26]. Once ingested by insect vector, RGDV establishes primary infection in the intestinal epithelial cells [27]. However, even in a single initially infected epithelial cell, the invading viruses initiate the formation of nascent viroplasms, which serve as the sites for viral multiplication to assemble progeny virions [27]. The progeny virions then pass directly through the basal lamina of the intestinal epithelium toward the visceral muscles, or cross the intercellular junctional complexes for cell-to-cell spread [27]. Viruses then disseminate from the visceral muscles into the hemolymphs and eventually into the salivary glands to be horizontally transmitted to healthy plants [27]. More importantly, RGDV can hitchhike with insect sperm for paternal transmission [20]. It was previously reported that the persistent replication of RGDV in cultured *R. dorsalis* cells can trigger the autophagy process, as demonstrated by the appearance of obvious virus-containing double-membraned autophagosomes and the conversion of ATG8-I to ATG8-II [18]. Applying the autophagy inhibitor 3-methyladenine or silencing the expression of ATG5 significantly decreases viral infection, whereas applying the autophagy inducer rapamycin facilitates viral infection [18]. Such autophagosomes are able to mediate a nonlytic viral release from insect cells [18]. Here, we further demonstrate that RGDV could effectively hijack autophagosomes to overcome the membrane barriers in vector midgut or salivary gland. Furthermore, the nonstructural protein Pns11 of RGDV could directly recruit ATG5-ATG12 conjugation to induce the formation of autophagosomes for the transport of viral particles. These findings highlight how RGDV can block autophagosome degradation to benefit viral propagation in its insect vector.

## Results

### RGDV infection induces incomplete autophagy

Transmission electron microscopy showed the appearance of double-membraned vesicles, which are morphologically typical autophagic vacuoles (autophagosomes) in the cytoplasm of RGDV-infected midgut epithelium of *R. dorsalis*, but not in that of nonviruliferous *R. dorsalis* (Fig 1A and 1B). Most of the observed autophagosomes contained viral particles (Fig 1A and 1B). Immunoelectron microscopy showed that viral particles within the autophagosomes were reacted with the RGDV P8 antibody (Fig 1C). Furthermore, the ATG8 antibody also specifically labeled such autophagosomes (Figs 1D and S1A). Thus, RGDV particles can be engulfed by virus-induced autophagosomes in *R. dorsalis*.

One hallmark of autophagy induction is the lipidation of ATG8 [28]. Autophagic adapter p62 is an indicator to assess autophagic flux, as it targets ATG8 and is also specifically degraded by the autophagic-lysosomal pathway [4]. RT-qPCR assay showed that the transcript levels of ATG5, ATG8 and ATG12 were significantly increased, while that of p62 and Lamp1 remained

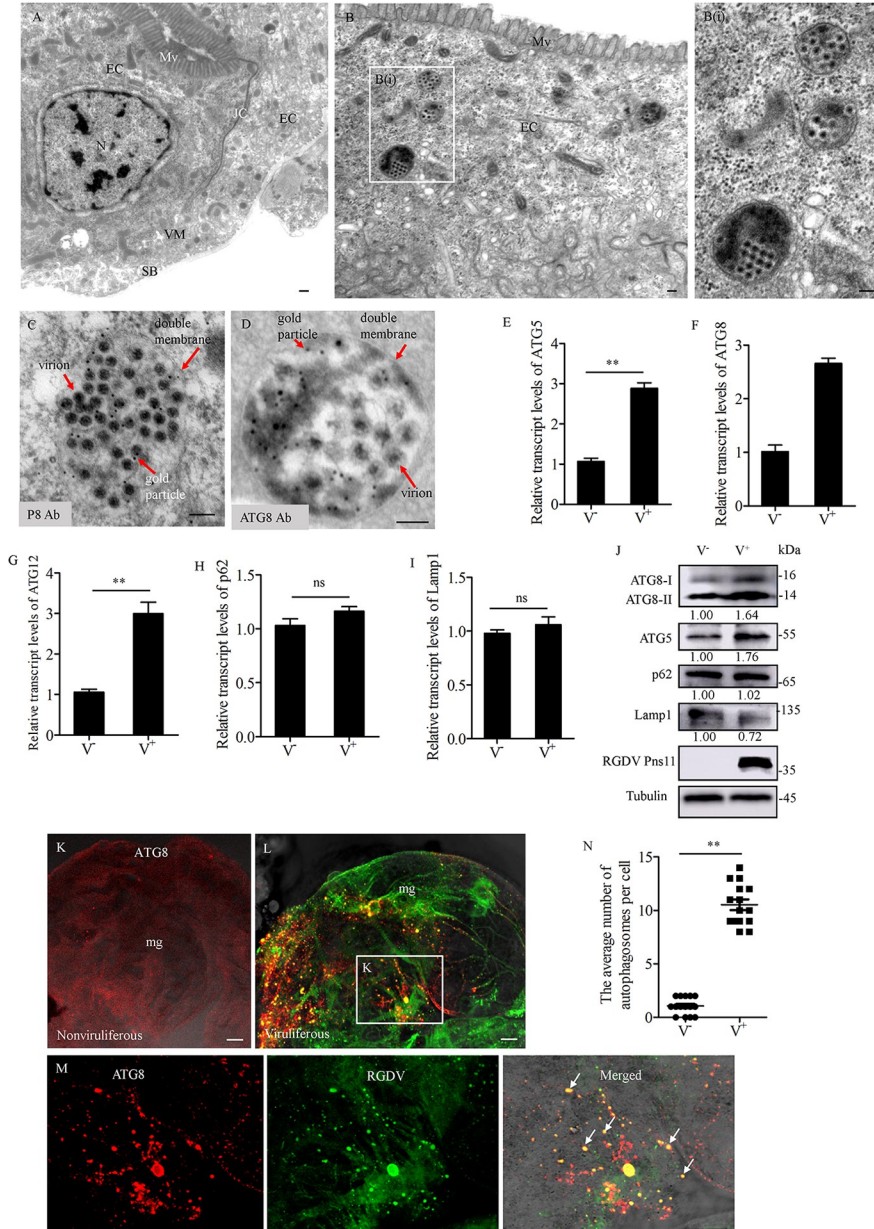

**Fig 1. RGDV infection induces autophagosome formation in the intestines of *R. dorsalis*.** (**A-D**) Transmission electron micrographs showing virus-induced autophagosomes in the midgut epithelial cells. (**A**) The midgut epithelium, showing microvilli, intercellular tight junctions, visceral muscles, and serosal barrier. (**B**) Virus-containing autophagosomes within virus-infected midgut epithelium. Panel **B** (I) is an enlargement of the boxed area in panel **B**. EC, epithelial cell; JC, junctional complexes; N, nucleus; Mv, microvilli; VM, visceral muscles; SB, serosal barrier. (**C, D**) Immunogold labeling of RGDV P8 (**C**) or ATG8 (**D**) in virus-containing autophagosomes. Virus-infected intestines of *R. dorsalis* were immunolabeled with P8- (**C**) or ATG8- (**D**) specific IgG as the primary antibody, followed by treatment with 10-nm gold particle-conjugated IgG as the secondary antibody. (**E-I**) Relative transcript levels for ATG5 (**E**), ATG8 (**F**), ATG12 (**G**), p62 (**H**) and Lamp1 (**I**) genes in nonviruliferous or viruliferous insects, as detected by RT-qPCR assay. Bars represent means ±SD from three independent experiments. Significance (**) was determined at $P < 0.01$. ns, not significant. (**J**) Relative protein expression levels of ATG8, ATG5, p62, Lamp1 and Pns11 in nonviruliferous or viruliferous insects as detected by western blot assay. Insect Tubulin was detected as a control. Relative protein levels were determined using ImageJ and the accumulation levels of ATG8-II, ATG5, p62, and Lamp1 in nonviruliferous insects were normalized to 1. (**K-M**) Immunofluorescence assay showing the distribution of ATG8-puncta in the midgut epithelial cells from nonviruliferous (**K**) or viruliferous (**L**) insects. The intestines were immunostained with P8-FITC (green) and ATG8-rhodamine (red). Panel **M** was an enlargement of the boxed area in panel **L** to show the colocalization of P8 and ATG8 (arrows). (**N**) The average number of ATG8-puncta per cell in

virus-free or infected midgut epithelial cells. Bars represent means ±SD from more than 10 individual cells. Significance (**) was determined at $P < 0.01$. V⁻, nonviruliferous. V⁺, viruliferous. Bars in A-D, 100 nm, Bars in K-M, 5 μm.

stable in viruliferous insects (Figs 1E–1I, and S2). Western blot assay showed that the intensity level of ATG8-II was increased in viruliferous insects relative to nonviruliferous controls (Figs 1J and S3A), suggesting that RGDV infection facilitates the conversion of ATG8-I to ATG8-II. Furthermore, the accumulation level of ATG5 was significantly increased, p62 remained stable, while N-glycosylated Lamp1 was significantly decreased in viruliferous insects (Fig 1J). Thus, the autophagosome-related p62 and ATG5 were not degraded after viral infection in *R. dorsalis.* Another hallmark of autophagy is the redistribution of ATG8 from a diffuse localization to a characteristic punctate pattern, reflecting the recruitment of ATG8 to autophagic vesicles [4]. Indeed, in virus-infected midgut epithelial cells, ATG8 redistributed into discrete puncta, which were co-localized with viral antigens (Fig 1K–1M). There was a significant increase ($P< 0.01$) in the average number of ATG8 puncta per cell in virus-infected midgut epithelial cells compared to uninfected cells (Fig 1N). Taken together, our results demonstrated that RGDV infection induced the incomplete autophagy within insect vectors.

## Nonstructural protein Pns11 of RGDV is a component of virus-induced autophagosomes

Because RGDV infection can induce the formation of autophagosomes in *R. dorsalis*, we further investigated if a single RGDV protein was responsible for this. Because intact viral particles were engulfed by virus-induced autophagosomes, immunoelectron microscopy was first employed to investigate which viral nonstructural proteins (Pns4, Pns7, Pns9, Pns10, Pns11 or Pns12) of RGDV localized to the autophagosomes. Immunoelectron microscopy showed that only Pns11 could specifically distribute in such autophagosomes, especially on the autophagosome membranes (Figs 2A and S1B). Previously, we reported that RGDV Pns11 induced the formation of tubular or fibrillar structures that facilitate viral dissemination within insect bodies [29,30]. Immunofluorescence microscopy further confirmed that RGDV Pns11 not only localized to the fibrillar structures, but also was associated with ATG8-containing puncta (Fig 2B), suggesting that Pns11 may be responsible for inducing autophagy in the insect vectors.

Previously, we have shown that virus-containing autophagosomes can directly fuse with the plasma membrane of cultured *R. dorsalis* cells to release viral particles [18]. We then used electron microscopy to investigate how virus-induced autophagosomes overcome insect transmission barriers. We observed that virus-containing autophagosomes could attach directly to the serosal side of the midgut epithelium barrier, and then release into the hemocoel (Fig 2C). Immunoelectron microscopy confirmed that the Pns11 antibody specifically labeled these autophagosomes (Fig 2C). We also observed that virus-containing autophagosomes could target and cross the intercellular junctional complexes between the adjacent midgut epithelial cells (Fig 2D). Furthermore, such autophagosomes also could target and traverse the apical plasmalemma into saliva-stored cavities in the salivary gland (Figs 2E and S1C). Taken together, RGDV can exploit autophagosomes as the vehicles to overcome the multiple membrane barriers, finally facilitating viral transmission by *R. dorsalis* (Fig 2F and 2G).

## RGDV Pns11 alone can induce autophagy

To confirm whether RGDV Pns11 alone can induce autophagy, we expressed Pns11 in *Spodoptera frugiperda* (Sf9) cells using recombinant baculovirus expression vector, and then

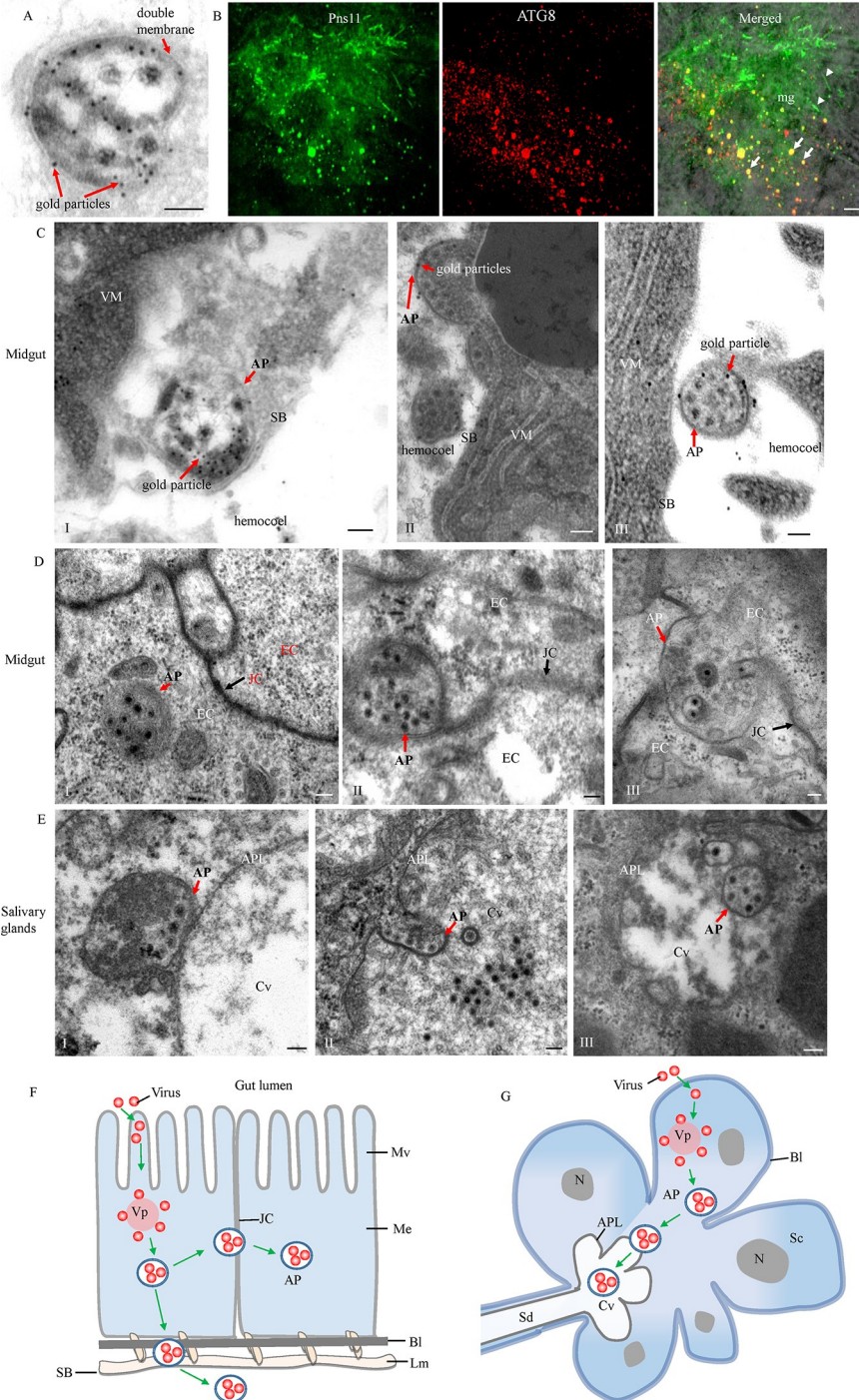

**Fig 2. RGDV exploits autophagosomes to overcome insect transmission barriers.** (**A**) Immunogold labeling of Pns11 in virus-containing autophagosomes. (**B**) At 6 days padp, insect intestines were immunolabeled with ATG8-rhodamine (red) and Pns11-FITC (green), and then examined by immunofluorescence microscopy. Arrows indicate the colocalization of Pns11 and ATG8 in the puncta. Arrowheads indicate the fibrillar structures of Pns11. (**C-E**) Transmission electron micrographs showing the exploitation of autophagosomes by RGDV to overcome membrane barriers in the midgut (**C** and **D**) and salivary gland (**E**). (**C**) Autophagosomes attached to the serosal barrier (panel I) and then released into the hemocoel (panels II and III). (**D**) Autophagosomes targeted (panels I and II) and crossed (panel III) the midgut intercellular junctional complexes. (**E**) Autophagosomes targeted (panel I) and traversed (panels II and III) the apical plasmalemma into saliva-stored cavities. Virus-infected intestines in **A** and **C** were immunolabeled with P11-specific IgG as the primary antibody, followed by treatment with 10-nm gold particle-

conjugated IgG as the secondary antibody. (**F, G**) Models for virus-induced autophagosomes to overcome the membrane barriers in vector midgut (**F**) and salivary gland (**G**). AP, autophagosome; APL, apical plasmalemma; Bl, basal lamina; Cv, cavity; EC, epithelial cell; JC, junctional complexes; Lm, longitudinal muscle; Me, midgut epithelium; Mv, microvilli; N, nucleus; Sc, salivary cytoplasm; SB, serosal barrier; Sd, salivary duct; VM, visceral muscle; Vp, viroplasm. Bars in A, C-E, 100 nm, Bar in B, 5 μm.

analyzed autophagy activity using western blot and immunofluorescence assays. In Sf9 cells co-expressing Pns11 and GFP-ATG8, western blot assay showed that the accumulation level of GFP-ATG8-I/GFP-ATG8-II was distinctly upregulated compared with that in Sf9 cells singly expressing GFP-ATG8 (Figs 3A and S3B). Furthermore, the expression of Pns11 alone in Sf9 cells also increased the accumulation level of ATG8-I/ATG8-II (Figs 3B and S3C). Immunofluorescence assay further showed that the average number of discrete puncta of ATG8 was remarkably higher when GFP-ATG8 and Pns11 were co-expressed than when GFP-ATG8 was expressed alone (Fig 3C, 3D and 3F), indicating that Pns11 alone can trigger the accumulation of autophagosomes. However, the co-expression of the major outer capsid protein P8 of RGDV with GFP-ATG8 failed to induce ATG8 puncta formation (Fig 3E). These data indicated that Pns11 alone was sufficient to induce autophagy *in vitro*.

## RGDV Pns11 directly interacts with ATG5

To determine the molecular mechanism by which Pns11 triggers autophagy, we firstly examined the roles of canonical autophagy factors in this process. We initially investigated which

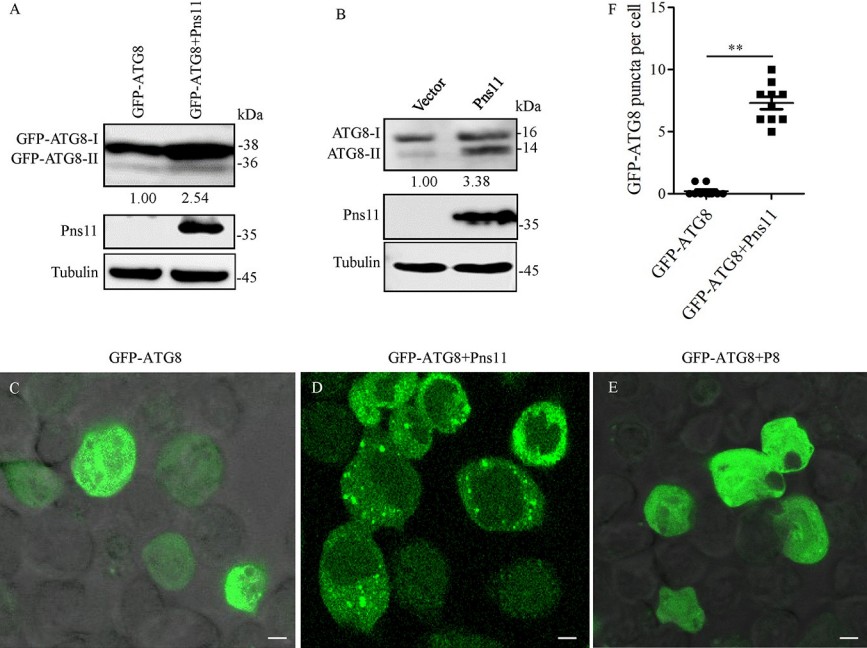

**Fig 3. RGDV Pns11 induces autophagy in Sf9 cells.** (**A**) Accumulation levels of ATG8 and RGDV Pns11 in Sf9 cells singly expressed with GFP-ATG8 or co-expressed with GFP-ATG8 and Pns11, as detected by western blot assay. Tubulin was detected as a control. (**B**) Accumulation levels of ATG8 and RGDV Pns11 in Sf9 cells infected with empty or recombinant bacmids expressing Pns11, as detected by western blot assay. Tubulin was detected as a control. Relative protein levels were determined using ImageJ and the accumulation levels of ATG8-II in the controls were normalized to 1. (**C-E**) Immunofluorescence assay showing GFP-ATG8 (green) in Sf9 cells singly expressed with GFP-ATG8, co-expressed with GFP-ATG8 and Pns11, or co-expressed with GFP-ATG8 and P8. (**F**) The average number of discrete puncta of GFP-ATG8 in Sf9 cells. Bars represent means ±SD from more than 10 individual cells. Significance (**) was determined at $P < 0.01$. Bars, 5 μm.

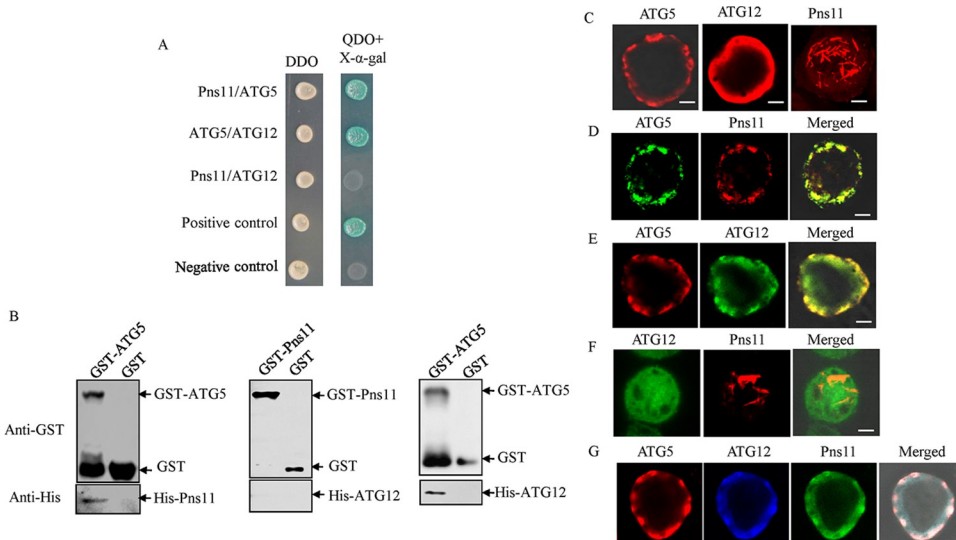

**Fig 4. Interactions among Pns11, ATG5 and ATG12.** (**A**) Interactions among Pns11, ATG5 and ATG12 were detected by yeast two-hybrid assay. Transformants were plated on either DDO or QDO+X-α-gal culture medium. Experiments were labeled as follows: Pns11/ATG5, pGBKT7-Pns11/pGADT7-ATG5; ATG5/ATG12, pGBKT7-ATG5/pGADT7-ATG12; Pns11/ATG12, pGBKT7-Pns11/pGADT7-ATG12; Positive control, pGBKT7-53/pGADT7-T; Negative control, pGBKT7-Lam/pGADT7-T. (**B**) The interactions among Pns11, ATG5 and ATG12 were detected by GST pull-down assay. GST-ATG5 or GST-Pns11 was incubated with glutathione-Sepharose beads. His-Pns11 or His-ATG12 was then added to the beads. Next, western blot assay was performed to detect His-Pns11 bound to GST-ATG5, His-ATG12 bound to GST-ATG5, and His-ATG12 bound to GST-Pns11. (**C-G**) Interactions among Pns11, ATG5 and ATG12 were detected in Sf9 cells. (**C**) At 48 hpi, Sf9 cells singly expressed with ATG5-His, ATG12-Flag or Pns11 were fixed and immunolabeled with His-rhodamine (red), Flag-rhodamine (red) or Pns11-rhodamine (red), respectively. (**D**) At 48 hpi, Sf9 cells co-expressed with ATG5-His and Pns11 were fixed and immunolabeled with His-FITC (green) and Pns11-rhodamine (red). (**E**) At 48 hpi, Sf9 cells co-expressed with ATG5-His and ATG12-Flag were fixed and immunolabeled with His-rhodamine (red) and Flag-FITC (green). (**F**) At 48 hpi, Sf9 cells co-expressed with Pns11 and ATG12-Flag were fixed and immunolabeled with Pns11-rhodamine (red) and Flag-FITC (green). (**G**) At 48 hpi, Sf9 cells triply-expressed with ATG5-His, ATG12-Flag and Pns11 were fixed and immunolabeled with Pns11-FITC (green), His-rhodamine (red), and Flag-Alexa Fluor 647 (blue). Bars, 5 μm.

autophagosome-related proteins (ATG3, ATG5, ATG7, ATG8, ATG10, and ATG12) from *R. dorsalis* could directly interact with Pns11 by yeast two-hybrid system. Of the six ATGs tested, only ATG5 interacted with Pns11 (Figs 4A and S4). Such interaction was independently verified using a glutathione S-transferase (GST) pull-down assay (Fig 4B). The formation of ATG5-ATG12 complex is necessary for phagophore membrane development [31]. Expectedly, ATG5 interacted with ATG12 from *R. dorsalis*, but Pns11 did not interact with ATG12 (Fig 4A and 4B).

Previously, we reported that Pns11 expression alone could induce the formation of fibrillar or tubular structures in Sf9 cells [29,32]. We then used the *in vitro* baculovirus expression system to investigate the relationship of Pns11, ATG5 and ATG12 in Sf9 cells. When expressed individually, Pns11 formed the fibrillar structures, ATG5 formed the punctate structures, and ATG12 remained diffuse throughout the cytoplasm of Sf9 cells (Fig 4C). Co-expression led to the redistribution of fibrillar structures of Pns11 into the punctate structures of ATG5, confirming the interaction between Pns11 and ATG5 (Fig 4D). Co-expression also led to the recruitment of ATG12 into the punctate structures of ATG5 (Fig 4E). Expectedly, Pns11 did not co-localize with ATG12 (Fig 4F). Finally, triple expression of Pns11, ATG5 and ATG12 in Sf9 cells resulted in the co-localization of all three proteins in the punctate structures (Fig 4G). Thus, RGDV Pns11 could be recruited by ATG5-ATG12 conjugation during autophagosome biogenesis via the direct interaction of ATG5 and Pns11.

## ATG5 is required for Pns11-induced autophagosome formation

We then used the RNA interference (RNAi) strategy to knock down the expression of ATG5, and then analyzed Pns11-induced autophagy using western blot and immunofluorescence assays. After feeding on RGDV-infected rice plants for 1 day, the second instar nymphs of *R. dorsalis* were microinjected with dsRNAs targeting ATG5 or GFP gene (dsATG5 or dsGFP). RT-qPCR and western blot assays showed that the accumulation levels of viral proteins, ATG5 and ATG8-I/ATG8-II were significantly decreased in dsATG5-treated insects as compared to dsGFP-treated controls (Figs 5A–5D and S3D). Immunofluorescence microscopy confirmed that the average number of the ATG8-Pns11 puncta in the midgut epithelial cells of dsATG5-treated insects was significantly reduced (Fig 5E and 5F). Accordingly, the transmission rate of dsATG5-treated insects was also significantly decreased (Fig 5K). Thus, ATG5 was required for RGDV-induced ATG8 lipidation, autophagosome formation and efficient viral infection and transmission in the insect vectors.

Consistent with these observations, we further microinjected the second instar nymphs of *R. dorsalis* with dsRNAs targeting Pns11 or GFP gene (dsPns11 or dsGFP) after feeding on RGDV-infected rice plants for 1 day. The knockdown of Pns11 expression effectively suppressed the accumulation levels of RDV P8, ATG5 and ATG8-I/ATG8-II, and the average number of ATG8-Pns11 puncta in the midgut epithelial cells of dsPns11-treated insects (Figs 5E–5J and S3E). Accordingly, the transmission rate of dsPns11-treated insects was also significantly decreased (Fig 5K). The results further indicated that the knockdown of Pns11 expression inhibited viral efficient propagation, which led to the failure of virus-induced autophagosome formation. Taken together, all the results suggest that Pns11 directly recruited ATG5-ATG12 conjugation to induce the formation of autophagosomes to facilitate viral dissemination within the insect vectors.

## RGDV Pns11 directly interacts with and mediates the reduced expression of N-glycosylated Lamp1 on lysosomal membranes

We have shown that viral infection led to the reduced expression of N-glycosylated Lamp1 on lysosomal membranes. Immunofluorescence microscopy showed that RGDV P8 and Pns11 were not associated with Lamp1-positive or LysoTracker-stained lysosomes in virus-infected midgut epithelial cells (Fig 6A–6D). There was a significant decrease ($P < 0.01$) in the average number of lysosomes per cell in virus-infected midgut epithelial cells compared to uninfected cells (Fig 6E). Thus, viral infection would mediate the degradation of lysosomes. Yeast two-hybrid assay showed that Pns11 directly interacted with Lamp1 from *R. dorsalis* (Fig 6F). Such interaction was independently verified using GST pull-down assay (Fig 6G). In Sf9 cells co-expressing Pns11 and Lamp1, western blot assay showed that the Lamp1 accumulation level was distinctly decreased compared with that in Sf9 cells singly expressing Lamp1 (Fig 6H). Thus, Pns11 alone could mediate the reduced expression of N-glycosylated Lamp1 on lysosomal membranes. RNAi assay was then performed to explore the role of Lamp1 during viral infection in *R. dorsalis*. RT-qPCR assay showed that the transcript levels of Pns11 and ATG5 in dsRNAs targeting Lamp1 gene (dsLamp1)-treated insects were significantly increased, while that of Lamp1 was significantly decreased as compared to dsGFP-treated controls (Fig 6I–6N). The knockdown of Lamp1 expression effectively increased the accumulation levels of Pns11, ATG5, p62, and ATG8-I/ATG8-II in dsLamp1-treated insects (Figs 6O and S3F), further confirming that the reduced expression of Lamp1 would benefit virus-induced autophagosome formation in viruliferous *R. dorsalis*. Taken together, all these results revealed that RGDV Pns11 potentially blocked the degradation of virus-containing autophagosomes by directly interacting with and mediating the reduced expression of N-glycosylated Lamp1 on lysosomal membranes.

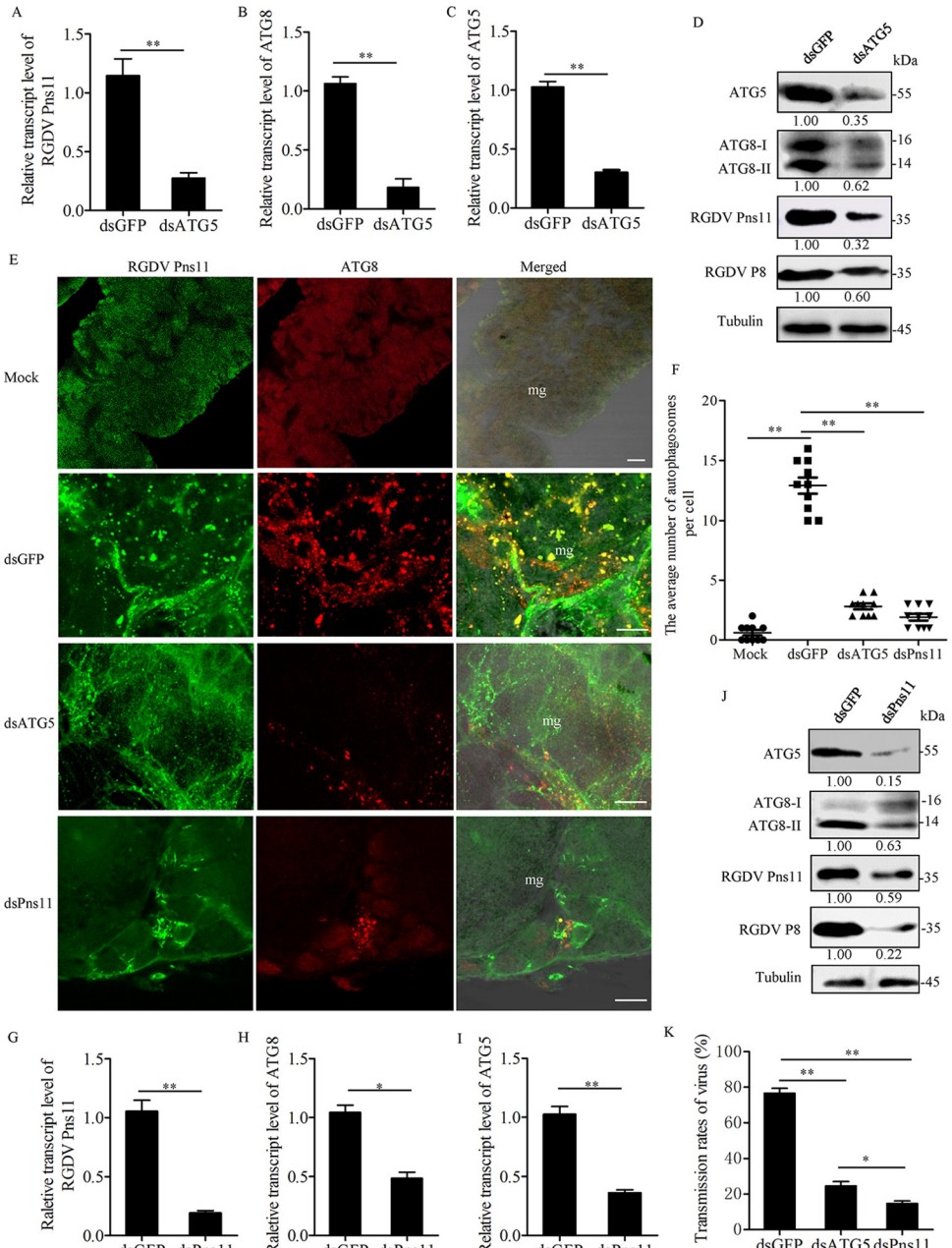

**Fig 5. ATG5 is required for Pns11-induced autophagosome formation.** (**A-C**) Relative transcript levels of Pns11 (**A**), ATG8 (**B**) and ATG5 (**C**) in viruliferous insects after dsGFP or dsATG5 treatment were measured by RT-qPCR assay. Bars represent means ±SD from three independent experiments. Significance (**) was determined at $P < 0.01$. (**D**) Accumulation levels of ATG8, ATG5, P8 and Pns11 in viruliferous insects after dsGFP or dsATG5 treatment were tested by western blot assay using ATG5-, ATG8-, P8- and Pns11-specific IgGs. Tubulin was detected as an internal control. Relative protein levels were determined using ImageJ and the accumulation levels of ATG8-II, ATG5, Pns11 and P8 in dsGFP-treated insects were normalized to 1. (**E**) The intestines of viruliferous insects after treatment of dsATG5, dsPns11 or dsGFP were immunolabeled with Pns11-FITC (green) and ATG8-rhodamine (red) at 6 days padp. Mg, midgut. Bars, 5 μm. (**F**) The average number of ATG8-puncta per cell of viruliferous insects after dsGFP, dsATG5 or dsPns11 treatment. Bars represent means ±SD from more than 10 individual cells. Significance (**) was determined at $P < 0.01$. (**G-I**) Relative transcript levels of Pns11 (**G**), ATG8 (**H**) and ATG5 (**I**) in viruliferous insects after dsGFP or dsPns11 treatment were measured by RT-qPCR assay. Bars represent means ±SD from three independent experiments. Significance (**) was determined at $P < 0.01$. (**J**) Accumulation levels of ATG8, ATG5, P8 and Pns11 in viruliferous insects after dsGFP or dsPns11 treatment, as determined by western blot assay using ATG5-, ATG8-, P8- and Pns11-specific IgGs. Tubulin was detected as an internal control. Relative protein levels were determined using ImageJ and the accumulation levels of ATG8-II, ATG5, Pns11 and P8 in dsGFP-treated insects were

normalized to 1. (**K**) Transmission rates of RGDV by individual viruliferous insects treated with different dsRNAs. Bars represent means ±SD from three independent experiments. Significance (\*\*) was determined at $P < 0.01$. Significance (\*) was determined at $P < 0.05$.

## Discussion

Viral pathogens have developed strategies to directly hijack autophagy for their own benefit during viral infection life [33]. Many positive-strand RNA viruses exploit the autophagic process for viral RNA replication through multiple mechanisms, including protecting viral RNAs from detection by innate immune sensors and degradation, stimulating protein translation, and generating energy or membrane structures required for viral replication [34]. Furthermore, several viruses, such as poliovirus, Hepatitis B virus, human parainfluenza virus type 3 (HPIV3), and influenza virus have been reported to hijack autophagosomes as a nonlytic mechanism for viral release from cultured mammalian cells [35–38]. However, whether autophagosome-mediated viral spread occurs in infected mammalian host tissues *in vivo* is unknown.

In general, arthropod-borne viruses establish their initial infection in the insect intestinal epithelium, disseminate to the hemolymph, and finally spread into the salivary glands, from which they are introduced into susceptible hosts together with saliva [39]. Two plant viruses, TYLCV and RBSDV, activate autophagy as an antiviral mechanism in the insect vectors; however, the virions were not directly enclosed within the autophagosomes [16,17]. RGDV virions are directly enclosed within virus-induced autophagosomes, and thus these autophagosomes can fuse with the plasma membranes for a nonlytic viral release from insect vector cells [18]. Here, we further demonstrate that RGDV hijacks such double-membraned autophagosomes to traverse the apical plasmalemma into saliva-stored salivary cavities, pass through the tight intercellular junctions into adjacent midgut epithelial cells, or release from the serosal barrier of midgut epithelium into the hemocoel. We thus determine that viruses can hijack autophagic release pathway to overcome the membrane barriers of insect organs. Viral nonstructural protein Pns11 could physically distribute on the membranes of virus-induced autophagosomes. RGDV Pns11 specifically interacts with cytoplasmic actin and actin-binding proteins such as myosin and gelsolin, which are the main components of insect membrane barriers [32,40]. Previously, we have shown that the actin filaments can provide sufficient power to propel the tubular or fibrillar structures constructed by RGDV Pns11 to spread from infected cultured leafhopper cells into adjacent cells [15,29]. Thus, we deduce that the attachment of Pns11 with cytoplasmic actin would also propel Pns11-associated double-membraned autophagic vesicles to pass through the tightly regulated membrane barriers of leafhopper vectors. Similarly, we have observed that viral particles of rice dwarf virus (RDV), also a phytoreovirus, were sequestered in the multivesicular bodies (MVBs) in the salivary glands of insect vectors [41]. Thus, RDV can hijack small vesicles, referred to as exosomes, to traverse the apical plasmalemma into saliva-stored cavities after fusing MVBs with apical plasmalemma [41]. In particular, exosome- or autophagosome-mediated viral spread through the membrane barriers does not cause substantial damage to insect tissues or organs, conferring an evolved advantage for viruses to be persistently transmitted by insect vectors. Furthermore, autophagic or exosomal vesicles potentially operate as an immune evasion strategy within insect bodies.

The expression of Pns11 increases both the conversion of ATG8-I to ATG8-II as well as the average number of GFP-ATG8 puncta in Sf9 cells. These results suggest that Pns11 alone is sufficient to induce autophagy. Additionally, among six ATG-related proteins, only ATG5 specifically interacts with Pns11. The co-expression of Pns11 and ATG5 leads to the redistribution of Pns11 into punctate structures of ATG5. Furthermore, Pns11, ATG5 and ATG12 co-localize to the punctate structures. Thus, Pns11 seems to be recruited by ATG5-ATG12 conjugation

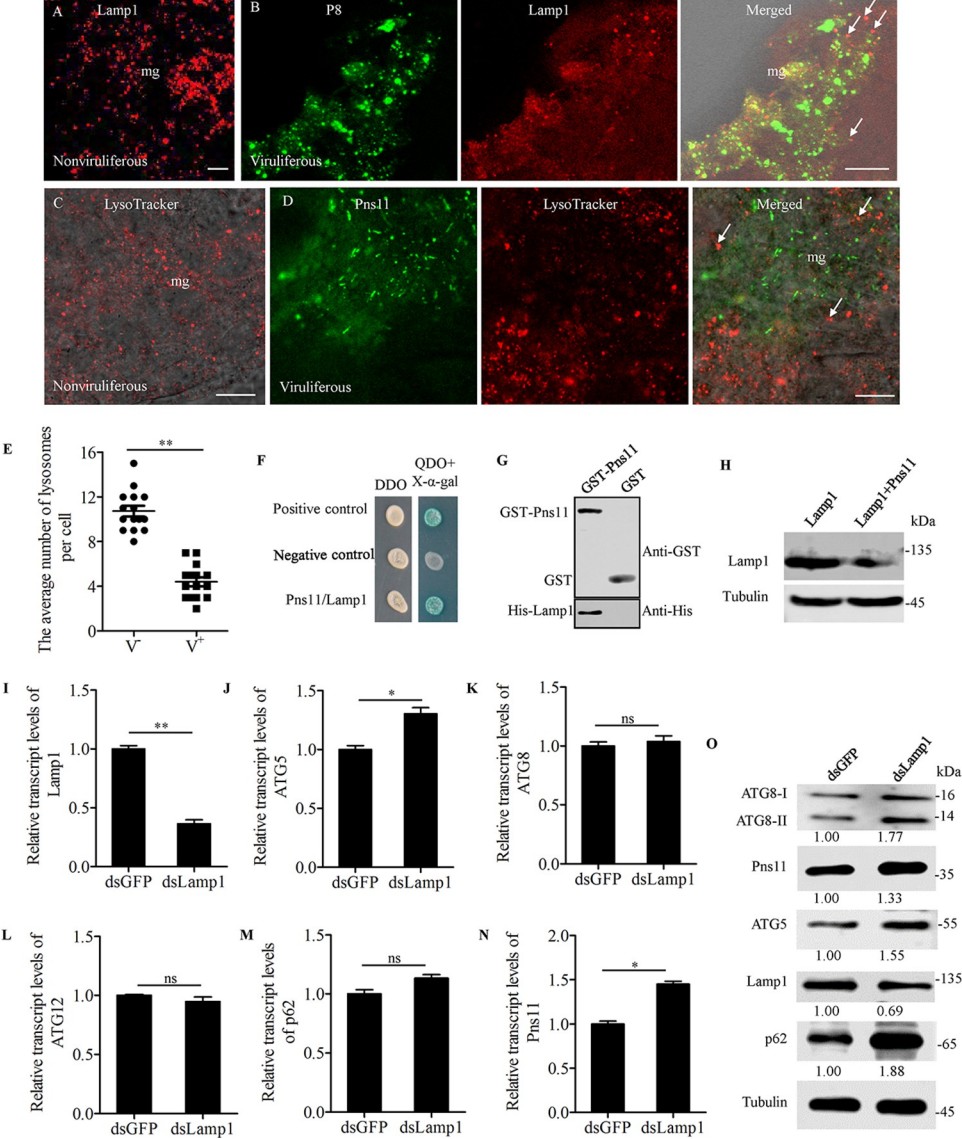

**Fig 6. Interaction between Pns11 and Lamp1.** (**A-D**) Immunofluorescence assay showing the distribution of lysosomes in the midgut epithelial cells from nonviruliferous (**A, C**) or viruliferous (**B, D**) insects. The intestines were immunostained with P8-FITC (green) and Lamp1-rhodamine (red) (**A, B**), or with Pns11-FITC (green) and LysoTracker (red) (**C, D**), and then examined by immunofluorescence microscopy. Arrows indicate lysosomes. Bars, 5 μm. (**E**) The average number of lysosomes per cell in virus-free (V⁻) or infected (V⁺) intestinal epithelial cells. Bars represent means ±SD from 15 individual cells. Significance (**) was determined at $P < 0.01$. (**F**) Interaction between Pns11 and LAMP1 was detected by yeast two-hybrid assay. Transformants on plates of either DDO or QDO+X-α-gal are shown. Experiments were labeled as follows: Positive control, pGBKT7-53/pGADT7-T; Negative control, pGBKT7-Lam/pGADT7-T; Pns11/Lamp1, pGBKT7-Pns11/pGADT7-Lamp1. (**G**) Interaction between Pns11 and Lamp1 was detected by GST pull-down assay. GST-Pns11 and GST were incubated with glutathione-Sepharose beads. His-Lamp1 was then added to the beads. Next, western blot assay was performed to detect His-Lamp1 bound to GST-Pns11. (**H**) Accumulation levels of Lamp1 in Sf9 cells singly expressed with Lamp1 or co-expressed with Lamp1 and Pns11, as detected by western blot assay. Tubulin was detected as a control. (**I-N**) Relative transcript levels of Lamp1 (**I**), ATG5 (**J**), ATG8 (**K**), ATG12 (**L**), p62 (**M**) and Pns11 (**N**) in viruliferous insects after dsGFP or dsLamp1 treatment were measured by RT-qPCR assay. Bars represent means ±SD from three independent experiments. Significance (**) was determined at $P < 0.01$. Significance (*) was determined at $P < 0.05$. ns, not significant. (**O**) Accumulation levels of ATG5, ATG8, Lamp1, p62 and Pns11 in viruliferous insects after treatment with dsGFP or dsLamp1, as determined by western blot assay using ATG5-, ATG8-, Lamp1-, p62- and Pns11-specific IgGs. Tubulin was detected as an internal control. Relative protein levels were determined using ImageJ and the accumulation levels of ATG8-II, ATG5, Pns11 and P8 in dsGFP-treated insects were normalized to 1.

during autophagosome biogenesis, via direct interaction of Pns11 and ATG5. The ATG5-ATG12 conjugation is essential for the proper curvature and elongation of the phagophore membranes [3]. Thus, it can be reasonably deduced that the binding of fibrillar Pns11 structures to the conjugated ATG5-ATG12 on the phagophore membranes may further remodel the double-membraned autophagosome to serve as a structural platform for viral accumulation and further nonlytic egress. A key event in the biogenesis of autophagosomes is the recruitment of ATG5-ATG12 conjugation to facilitate the covalent attachment of ATG8 to PE lipids on the nascent phagophore membranes [31]. It was further determined here that the ATG5-ATG12 conjugation is required for RGDV-induced ATG8 lipidation, the subsequent autophagosome formation and, ultimately, efficient viral dissemination throughout the insect vectors. ATG8 is present on both inner and outer membranes of the autophagosomes, and also serves as an adaptor for selective substrates such as p62 [42]. In general, the autophagosome-related ATG8 and p62 are efficiently degraded after the fusion of autophagosomes with lysosomes [43]. However, we observe here that such autophagic flux is inhibited during viral infection, potentially via blocking the fusion of virus-containing autophagosomes with lysosomes. The reduced expression of N-glycosylated Lamp1 in viruliferous *R. dorsalis* further supports the mechanism of action in this process. Others have reported that viral infection effectively disrupted the fusion of autophagosomes with lysosomes by interfering with the function of the regulators of autophagic flux [36,44]. For example, the NSP4 protein of rotavirus binds to autophagosomes and inhibits their fusion with lysosomes to enhance viral RNA replication [44]. Similarly, the phosphoprotein of HPIV3 disrupts the fusion of autophagosomes with lysosomes to promote extracellular viral production [36]. We further determine that Pns11 of RGDV could effectively mediate the reduced expression of N-glycosylated Lamp1 on lysosomal membranes by directly targeting Lamp1, finally decreasing the fusion ability of virus-containing autophagosomes with lysosomes by an unknown mechanism. Thus, RGDV Pns11 is involved in preventing the degradation of virus-induced autophagosomes. Taken together, we show that RGDV Pns11 induces a non-canonical autophagy process to benefit viral spread in the insect vectors.

In conclusion, we demonstrate that a vector-borne virus is able to exploit the autophagic constituents to form membranous structures implicated to be involved in viral spread within the regions that we have examined. The nonstructural protein Pns11 of RGDV induces the formation of double-membraned autophagosomes that are dependent on ATG5-Pns11 interaction. Such autophagosomes are modified to evade degradation, thus can be persistently exploited by RGDV to safely transport virions across multiple insect membrane barriers (Fig 7). It seems reasonable to speculate that this process may be a conserved strategy for vector-borne viruses to ensure that the viruses can be persistently transmitted by their insect vectors.

## Materials and methods

### Ethics statement

The pplyclonal antibodies against ATG5 peptide QLNPEEANKLQPPE, p62 peptide SSQSER-RESSPAAE, Lamp1 peptide DEPSKPNEKAASES and ATG8 peptide FLIRKRVHLRPEDA were produced in rabbit by Genscript USA Innovation Company (Nanjing), which is approved by the Science Technology Department of Jiangsu Province, China with approval number SYXK (Su) 2018–0015.

### Insects, viruses, cells and antibodies

Nonviruliferous individuals of *R. dorsalis* were collected from Guangdong Province, China and propagated at 25 ± 3˚C in the laboratory. The RGDV-infected rice plants were also

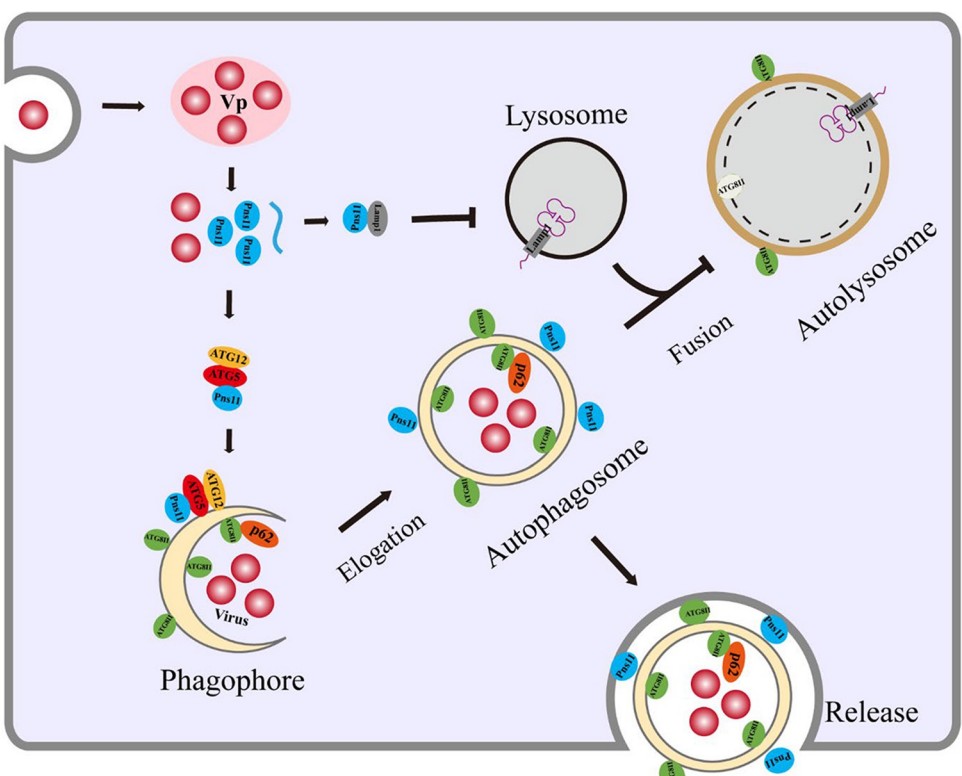

**Fig 7. A model depicting ATG5-ATG2 conjugation-dependent Pns11-induced autophagosome formation during RGDV infection in *R. dorsalis*.** After virions enter the midgut epithelium of *R. dorsalis* and initiate multiplication process for the assembly of progeny virions, viral nonstructural protein Pns11 is initially synthesized and recruited by ATG5-ATG12 conjugation on the phagophore membranes. This process then induces the formation of double-membraned autophagosomes for packaging of virions. Meanwhile, RGDV Pns11 directly interacts with and mediates the reduced expression of N-glycosylated Lamp1 on the lysosomal membranes, which potentially preventing the fusion of Pns11-associated autophagosomes with lysosomes. Autophagosomes can traverse multiple membrane barriers of insect vector to enhance viral spread.

collected from Guangdong Province, China, and propagated via transmission by *R. dorsalis* as reported previously [45]. Sf9 cells were cultured and maintained in LBM growth medium [29]. Antibodies against major outer capsid protein P8 and nonstructural protein Pns11 of RGDV were prepared as described previously [46]. Polypeptides of ATG5 (QLNPEEANKLQPPE), p62 (SSQSERRESSPAAE), Lamp1 (DEPSKPNEKAASES) and ATG8 (FLIRKRVHLRPEDA) from *R. dorsalis* were conjugated to Keyhole Limpet Hemocyanin (KLH), then injected into rabbits to generate antibodies against the respective peptide antigens. These antibodies were produced by Genscript USA Innovation Company (Nanjing). IgGs were isolated from specific polyclonal antisera using a protein A-Sepharose affinity column (Pierce). The antibodies were then directly conjugated with FITC or rhodamine according to the manufacturer's instructions (Invitrogen). Alexa Fluor 488 rabbit monoclonal antibody against 6 x His tag (ab237336) and Alexa Fluor 488 rabbit monoclonal antibody against Flag tag (ab245892) were obtained from Abcam Company.

## Detection of RGDV or *R. dorsalis* gene expression by RT-qPCR assay

To determine the effects of RGDV infection on the expression of genes associated with autophagy in *R. dorsalis*, total RNAs of 50 intact insects were extracted using Trizol Kit (Invitrogen) following the manufacturer's instructions. Reverse transcription was then performed using

RevertAid reverse transcriptase (Invitrogen) and the appropriate reverse primers. The primer sequences were listed in S1 Table. RT-qPCR assay was performed in triplicate using the SYBR Green PCR Master Mix kit (GenStar) according to the manufacturer's instructions. The elongation factor 1 (EF1) transcript of *R. dorsalis* was used to quantify the relative transcript levels. Relative gene expression levels were calculated using the $2^{-\Delta\Delta CT}$ method [47].

## Detection of viral and insect protein expression by western blot assay

We then investigated the effects of RGDV infection on viral or insect protein expression. Briefly, 50 insects were ground in liquid nitrogen and then dissolved with 0.2 mL lysis buffer (0.5 M phosphate, 0.5 M NaCl, 0.1% Tween-20, 0.1% NP-40, 1 mM phenylmethanesulfonyl fluoride, 0.1% β-mercaptoethanol, 10 mM Tris-HCl, pH 8.0) at 4˚C. After centrifugation (12,000 rpm, 20 min at 4˚C), the supernatant was transferred to a new tube, and the protein concentration was measured using the Bradford assay. Protein samples were separated by SDS-PAGE and transferred onto polyvinylidene difluoride (PVDF) membranes. After blocking with 5% nonfat milk, the membranes were incubated with P8-, Pns11-, ATG5-, ATG8-, p62-, Lamp1- and Tubulin-specific IgGs served as the primary antibodies, and goat anti-rabbit IgG-peroxidase (Sigma) served as the secondary antibody. The bands were detected using the Luminata Classico Western HRP Substrate (Millipore) and imaged with the Molecular Imager ChemiDoc XRS+ System (Bio-Rad).

## Examination of autophagy or lysosome by immunofluorescence microscopy

To observe the autophagy during viral infection in *R. dorsalis*, 200 second instar nymphs were fed on RGDV-infected rice plants for 1 day, and then placed on healthy rice plants. At different days post-first access to diseased plants (padp), the intestines of 30 insects were fixed in 4% paraformaldehyde for 2 h and permeabilized in 0.2% Triton X-100 in 0.01 M PBS buffer for 30 min at room temperature. The internal organs were immunolabeled with ATG8- or Lamp1-specific IgG conjugated to rhodamine (ATG8-rhodamine or Lamp1-rhodamine), P8- or Pns11-specific IgG conjugated to FITC (P8-FITC or Pns11-FITC), and then processed for immunofluorescence microscopy. The intestines dissected from *R. dorsalis* adults fed on healthy rice plants were treated exactly the same way and served as controls. Meanwhile, the virus-infected or uninfected intestines were stained with the lysosome dye LysoTracker (Thermo Fisher Scientific) for 30 min according to the manufacturer's protocol. The stained samples were fixed, permeabilized, immunolabeled with Pns11-FITC, and then processed for immunofluorescence microscopy.

## Electron microscopy

The intestines and salivary glands dissected from nonviruliferous or viruliferous *R. dorsalis* adults were fixed, dehydrated, and embedded, and the ultrathin sections were cut as previously described [30]. For immunoelectron microscopy, the sections were immunolabeled with P8-, Pns11- or ATG8-specific IgG as the primary antibody, followed by treatment with goat anti-rabbit IgG conjugated with 10-nm-diameter gold particles as the secondary antibody (Abcam). Thin sections were examined with an H-7650 Hitachi transmission electron microscope (Hitachi, Tokyo, Japan).

## Yeast two-hybrid assay

Yeast two-hybrid screening for the interactions between Pns11 and autophagy- or lysosome-related proteins (ATG3, ATG5, ATG7, ATG8, ATG10, ATG12 or Lamp1) was performed

using the Matchmaker Gal4 Two-Hybrid System3 (Clontech) according to the manufacturer's protocol. The full-length ORFs of ATG3, ATG5, ATG7, ATG8, ATG10, ATG12 or Lamp1 from *R. dorsalis*, as well as Pns11 of RGDV were amplified and cloned into either the bait (pGBKT7) or prey (pGADT7) plasmids, respectively. The bait and prey plasmids were used to co-transfect yeast strain AH109, and β-galactosidase activity was detected on SD/Leu-Trp-His-Ade-/X-a-gal culture medium. The positive control pGBKT7-53/pGADT7-T and negative control pGBKT7-Lam/pGADT7-T were transfected in the same way.

## Baculovirus expression assay

The baculovirus system was used to express ATG5, ATG12 or Pns11 for investigating their relationships in Sf9 cells. Briefly, the full-length ORFs of ATG5 tagged with His in the C-terminal (ATG5-His), ATG12 tagged with Flag in the C-terminal (ATG12-Flag), and Pns11 were cloned into the baculovirus vector pFastBac1 using the ClonExpress II one step cloning kit (Vazyme). Subsequently, the recombinant baculovirus vectors were introduced into *E. coli* DH10Bac strain to generate the recombinant bacmids containing ATG5-His, ATG12-Flag and Pns11, which were then singly or co-transfected into Sf9 cells using the Cellfectin II Reagent (Invitrogen). At 48 h post-infection (hpi), Sf9 cells were fixed with 4% paraformaldehyde, immunolabeled with His-, Flag- or Pns11-specific IgG conjugated to FITC, rhodamine, or Alexa Fluor 647 (Invitrogen), and then processed for immunofluorescence microscopy, as described previously [48].

To observe whether the expression of Pns11 could trigger autophagy in Sf9 cells, the baculovirus system was used to express RGDV P8 or ATG8 fused with GFP in the N-terminal (GFP-ATG8). Recombinant bacmids containing GFP-ATG8 or P8 were generated as described above. Recombinant bacmids expressing GFP-ATG8 were co-transfected with the recombinant bacmids expressing Pns11 or P8 into Sf9 cells. After 48 h, the cells were visualized using a Leica TCS SP5 inverted confocal microscope. Furthermore, the ATG8-I/ATG8-II conversion in Sf9 cells co-expressed with GFP-ATG8 and Pns11, or singly expressed with Pns11, was tested by western blot assay with ATG8 antibody. To observe whether the expression of Pns11 could mediate the down-regulation of Lamp1 expression in Sf9 cells, the baculovirus system was used to express Lamp1. Recombinant bacmids containing Lamp1 were co-transfected with the recombinant bacmids expressing Pns11 into Sf9 cells. The accumulation levels of Lamp1 in Sf9 cells co-expressed with Lamp1 and Pns11, or singly expressed with Lamp1, were tested by western blot assay with the Lamp1 antibody.

## GST pull-down assay

The ORFs of Pns11 or ATG5 from *R. dorsalis* were cloned into pGEX-4T for fusion with GST tag. The ORFs of ATG12 or Lamp1 from *R. dorsalis* and Pns11 were cloned into pDEST17 for fusion with His tag. All recombinant proteins were expressed in *E. coli* Rosetta strain and purified. The GST-Pns11 or GST-ATG5 fusion proteins were first bound to GST-Sepharose 4B beads (GE) for 3 h at 4˚C. The mixture was then centrifuged for 5 min at 100 g, and the supernatant was discarded. His-ATG12, His-Pns11 and His-Lamp1 were then added to the beads, and incubated for 2 h at 4˚C. After 3 rounds of centrifugation and washing, the bead-bound proteins were separated by SDS-PAGE and detected by western blot assay using anti-His and anti-GST antibodies (Sigma).

## Knocking down ATG5, Pns11 or Lamp1 expression

We then used RNAi strategy to determine the functional roles of ATG5, Pns11 or Lamp1 during viral infection in *R. dorsalis*. Briefly, dsRNAs targeting 500 bp regions of ATG5, Pns11,

Lamp1 or GFP genes were synthesized *in vitro* using the T7 RiboMAX Express RNAi System (Promega). Two hundred second instar nymphs were microinjected with 0.5 μg/μl dsRNAs targeting their respective sequences (dsATG5, dsPns11, dsLamp1 or dsGFP) after feeding on RGDV-infected rice plants for 1 day. The insects were then fed on healthy rice seedlings for 5 days. The internal organs of RGDV-infected *R. dorsalis* adults were dissected, fixed, immuno-labeled with Pns11-FITC and ATG8-rhodamine, and then processed for immunofluorescence microscopy. The relative gene expression levels of ATG5, ATG8, ATG12, p62, Lamp1 and Pns11 were detected by RT-qPCR assay. Relative protein expression levels of ATG5, ATG8, p62, Lamp1, P8 and Pns11 were detected by western blot assay.

## Effects of dsATG5 or dsPns11 treatment on viral transmission by *R. dorsalis*

To investigate the transmission rates of RGDV by *R. dorsalis* individuals treated with dsATG5 or dsPns11, more than 300 second-instar nymphs were fed on RGDV-infected rice plants for 1 day, and then 100 insects were microinjected with dsATG5, dsPns11 or dsGFP, respectively. At 10 days padp, the individual insects were placed in the glass tubes containing the rice seed-lings. The insects were maintained for 10 days, with the seedlings replaced daily, as described previously [18]. Thirty insects and three replicates were conducted for the tests. The insects were collected and analyzed by RT-PCR assay at 20 days padp. The plants inoculated with the confirmed viruliferous insects were subjected to RT-PCR detection 10 days later. The trans-mission rates of RGDV were calculated as the percentage of RT-PCR positive plants out of the total number of plants [49].

## Statistical analyses

All data were analysed with SPSS, version 17.0. Multiple comparisons of the means were con-ducted based on Tukey's honest significant difference (HSD) test using a one-way analysis of variance (ANOVA). The data were back-transformed after analysis for presentation in the text and figures.

## Supporting information

**S1 Fig. The autophagic vesicles appear in viruliferous *R. dorsalis*.** (**A-C**) Immunogold label-ing of ATG8 (**A**), Pns11 (**B**) or P8 (**C**) in virus-containing autophagosomes in the midgut (**A, B**) or salivary gland (**C**). Virus-infected internal organs were immunolabeled with ATG8- (**A**), Pns11- (**B**) or P8- (**C**) specific IgG as the primary antibody, followed by treatment with 10-nm gold particle-conjugated IgG as the secondary antibody. Cv, cavity. Red arrows indicate gold particles. Bars, 100 nm.
(TIF)

**S2 Fig. Relative transcript levels of different ATG proteins in viruliferous or nonvirulifer-ous *R. dorsalis*.** The relative transcript levels of ATG1 (**A**), ATG3 (**B**), ATG6 (**C**), ATG7 (**D**), ATG9 (**E**) and ATG10 (**F**) were detected by RT-qPCR assay. Data are presented as means ±SD from three independent experiments. Significance (**) was determined at $P < 0.01$. ns, not sig-nificant.
(TIF)

**S3 Fig. The conversion of ATG8-I to ATG8-II in different treatments.** Relative accumula-tion levels were determined using ImageJ, and the ratios of ATG8-II to ATG8-I or Tubulin in Figs. 1J (**A**), 3A (**B**), 3B (**C**), 5D (**D**), 5J (**E**) and 6L (**F**) were analyzed. Significance (*) was determined at $P < 0.05$. Significance (**) was determined at $P < 0.01$. Significance (***) was

determined at $P < 0.001$. ns, not significant.
(TIF)

**S4 Fig. Yeast two hybrid assay for detection of the interactions between Pns11 and other ATG proteins of _R. dorsalis_.** Interactions between Pns11 and ATG3, ATG7, ATG8 or ATG10 were detected by yeast two-hybrid assay. Transformants were plated on either DDO or QDO+X-α-Gal culture medium, and the pairs are labeled as follows: Pns11/ATG3, pGBKT7-Pns11/pGADT7-ATG3; Pns11/ATG7, pGBKT7-Pns11/pGADT7-ATG7; Pns11/ ATG8, pGBKT7-Pns11/pGADT7-ATG8; Pns11/ATG10, pGBKT7-Pns11/pGADT7-ATG10; Positive control, pGBKT7-53/pGADT7-T; Negative control, pGBKT7-Lam/pGADT7-T.
(TIF)

**S1 Table. Primers used in this study.**
(DOC)

## Author Contributions

**Conceptualization:** Taiyun Wei.

**Data curation:** Dongsheng Jia, Qifu Liang, Huan Liu, Guangjun Li.

**Formal analysis:** Taiyun Wei.

**Funding acquisition:** Dongsheng Jia, Taiyun Wei.

**Investigation:** Guangjun Li, Xiaofeng Zhang, Qian Chen.

**Methodology:** Dongsheng Jia, Qifu Liang, Taiyun Wei.

**Validation:** Qifu Liang, Huan Liu.

**Visualization:** Dongsheng Jia.

**Writing – original draft:** Dongsheng Jia, Taiyun Wei.

**Writing – review & editing:** Aiming Wang, Taiyun Wei.

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
