## [Decision Letter · Decision Letter 0]

2 Jan 2022

Dear Dr Wei,

Thank you very much for submitting your manuscript "A nonstructural protein encoded by a rice reovirus induces an incomplete autophagy to promote viral spread in insect vectors" for consideration at PLOS Pathogens. As with all papers reviewed by the journal, your manuscript was reviewed by members of the editorial board and by several independent reviewers. In light of the reviews (below this email), we would like to invite the resubmission of a significantly-revised version that takes into account the reviewers' comments.

All reviewers note that new information is presented in this manuscript, but all reviewers raise important concerns. I agree with many of their concerns.

Reviewer 1 raises an important point about interpretation of autophagosome delivery of virus particles. This is based on your TEM data and needs to be clarified.

Reviewer 2 recommends major revision, mostly regarding the specific language used in the manuscript and the detail/clarity of the figure labeling. Some examples are given.

Reviewer 3 recommends rejection, and raises a number of important points which must be addressed. The reviewer raises the points in Figures 1G and 1H about Lamp1 reductions. The RT-qPCR data in 1G do not support this, and 1H shows no statistics to document the claim. See your comments on lines 156 and 157. This reviewer also raises the point concerning statements about Lamp1 on the lysosomal membrane on Line 257.

You also make global statements about the significance of your findings here with other insect vectored plant viruses, for example see the last line of the abstract and even the last line of the discussion suggests that this is a conserved property. This is over-reaching what you present here.

We cannot make any decision about publication until we have seen the revised manuscript and your response to the reviewers' comments. Your revised manuscript is also likely to be sent to reviewers for further evaluation.

Sincerely,

Bryce W. Falk

Guest Editor

PLOS Pathogens

Peter Nagy

Section Editor

PLOS Pathogens

Kasturi Haldar

Editor-in-Chief

PLOS Pathogens

orcid.org/0000-0001-5065-158X

Michael Malim

Editor-in-Chief

PLOS Pathogens

orcid.org/0000-0002-7699-2064

All reviewers note that new information is presented in this manuscript, but all reviewers raise important concerns. I agree with many of their concerns.

Reviewer 1 raises an important point about interpretation of autophagosome delivery of virus particles. This is based on your TEM data and needs to be clarified.

Reviewer 2 recommends major revision, mostly regarding the specific language used in the manuscript and the detail/clarity of the figure labeling. Some examples are given.

Reviewer 3 recommends rejection, and raises a number of important points which must be addressed. The reviewer raises the points in Figures 1G and 1H about Lamp1 reductions. The RT-qPCR data in 1G do not support this, and 1H shows no statistics to document the claim. See your comments on lines 156 and 157. This reviewer also raises the point concerning statements about Lamp1 on the lysosomal membrane on Line 257.

You also make global statements about the significance of your findings here with other insect vectored plant viruses, for example see the last line of the abstract and even the last line of the discussion suggests that this is a conserved property. This is over-reaching what you present here.

Reviewer's Responses to Questions

**Part I - Summary**

Reviewer #1: The manuscript is about inducing an incomplete autophagy by the rice gall dwarf virus to increase virus spread in its insect vector. Autophagy is considered as an important defense mechanism against pathogens evading, and how viruses have evolved to subvert this mechanism for their benefit is an interesting topic in the field of virology. This fact that viruses target the autophagy as a defense mechanism of host cells is not new and it has been shown for some animal and plant viruses. However, most of these results were illustrated in vitro. This manuscript shows this phenomenon in vivo which is interesting. The authors showed that one of the viral non-structural proteins (Pns11) is involved in formation of autophagosomes by interacting with ATG5. Additionally, Pns11 downregulates the expression of Lamp1 to prevent degradation of autophagosomes by lysosome. The manuscript written well, and results presented nicely.

My main question/concern is about the conclusion/claim that the authors has made about the role of autophagosomes in virus spread by the data presented in this manuscript. Although the EM shows that viral particles associated with autophagosomes are released from mid gut to hemocoel and the presence of the autophagosomes that contain viral particles was also showed in salivary glands, I am concerning/wondering if this is enough to conclude and claim that autophagosomes facilitate the virus transportation across insect membrane barriers. This is a strong statement. Do authors have any plans to use or suggest any other methods to validate this conclusion/claim?

Reviewer #2: This study described the discovery and characterization of enzymatically inactive autophagosomes induced by a virus (rice gall dwarf reovirus) in the cells of the insect that is both a host of this virus, and a transmitting vector. These “incomplete” autophagosomes were incapable of degrading the virus particles they enclosed, but were instead repurposed by the virus for its own spread inside the insect. The authors established that such autophagosomes were induced by nsP11, one of the RGDV-encoded proteins, by its direct interaction with ATG8. Furthermore, inhibiting their formation by down-regulating nsP11 or ATG8 expression compromised RGDV infection of the insect.

Overall the manuscript is reasonably well written, and the data support the conclusions drawn. The quality of the figures is generally good but better labeling could improve the readability (see below).

The language needs to be more carefully proof-read. Numerous examples are provided below, though they are by no means comprehensive.

Reviewer #3: The manuscript by Jia et al provides evidence to demonstrate the role of a nonstructural protein, Pns11, of the rice gall dwarf virus (RGDV) in controlling autophagosome formation in leafhoppers through its association with an autophagy-related protein ATG5, and the apparent correlation between this interaction and the conversion of the ATG8-I to ATG8-II complex (a feature key to the formation of autophagosomes). The aforesaid interaction also is shown to be associated with another autophagy-related protein ATG12, an autophagic adapter protein p62, and a lysosomal protein Lamp1. The manuscript also presents results showing the direct interaction between RGDV Pns11 and Lamp1 and the lack of interaction between RGDV P8 (the outer capsid protein) and Lamp1 as evidence that the RGDV Pns11-Lamp1 interaction prevents the formation of autolysosomes and degradation of RGDV.

The manuscript shows interesting associations among RGDV Pns11, ATG5, p62 and Lamp1 at the molecular level, and implicates the role of RGDV Pns11 in eliciting changes in the ATG8 complexes. The TEM micrographs showing the presence of RGDV in double-membraned autophagosomes within the midgut and salivary glands are conspicuous and of high quality. However, there are several technical gaps in the studies that give some pause to the conclusions made, indicating that the conclusions may be too preliminary. The manuscript also lacks important background information and is scattered with many reporting and editorial errors on top of missing information in specific sections of the results.

**Part II – Major Issues: Key Experiments Required for Acceptance**

Reviewer #1: (No Response)

Reviewer #2: None.

Reviewer #3: 1) Introduction: There needs to be some more background information about phagosome formation with details describing what is known (or unknown) about phagosome formation in animal and insect systems. Questions such as ‘how is ATG8-II related to ATG8-I?’ and ‘what is the basis for the difference in Mr between ATG8-II and ATG8-I as seen in many of the figures (e.g. Figs. 1H, 2B, 3A and 3B, 5B, 5J)?’ need to be addressed with information. Information about the viral encoded proteins is also needed since tests were done to determine if several non-structural proteins of the virus interacted with autophagosomes (L171). What functions are these ns proteins involved in? Another example of the lack of background information is that of the structural protein RGDV P8. Anti-P8 IgGs (Materials and Methods L402) were used to localize RGDV in the intestines (e.g. in Fig. 1K); why is the presence of P8 representative of the virus? One has to reach as far in as the third result section (corresponding to the results in Fig. 3) to find out that P8 is the RGDV outer capsid protein.

2) Results associated with Fig. 1: it is mentioned in the introduction that the coupling of ATG12 and ATG5 are involved in heterodimer formation on the phagophore. Therefore, it makes sense that the level of ATG12 also ought to be determined in viruliferous and non-viruliferous insects. However, this was not done, and no explanation was provided.

3) Results in Fig. 1H are consistent with the writing in L149-L151, but there is no indication here or in L377 (M&M) if quantified, equal amounts of total RNA was used for the samples. Neither was there an indication of how many insects were used for total RNA extraction (in M&M).

4) The experimental results describing the increased accumulation of GFP-ATG8-II in SF9 cells co-transfected with GFP-ATG8 and Pns11 or with GFP-ATG8 alone (L198-L199, Fig. 3A) are not convincing. To begin with, there appears to be more GFP-ATG8-I in SF9 cells co-transfected with GFP-ATG8 and Pns11 than in SF9 cells transfected with GFP-ATG8. It would be more appropriate to determine the ratio of GFP-ATG8-I to GFP-ATG8-II in SF9 cells expressing Pns11 and GFP-ATG8 vs that of SF9 cells expressing GFP-ATG8 alone.

5) The Y2H and pull-down assays in Fig. 5 clearly show the association between Pns11 and Lamp1. However, one cannot rule out that within the midgut epithelia, the interaction could well be occurring at multiple locations in the cell cytoplasm and not just in the lysosomal membrane. Therefore, this ought to be tested in an experiment to determine if Pns11 and Lamp1 are co-localized in the lysosomes within the midgut epithelia as well as in the salivary glands of viruliferous insects. It is also unclear what Fig. 6E is meant to convey. It seems the conclusion drawn from the results is that the interaction of Pns11 and Lamp1 in SF9 cells leads to the degradation of Lamp1; hence, by inference, the same phenomenon is also occurring in the midgut of viruliferous leafhoppers. This conclusion is too presumptive. To be able to make a solid conclusion that the association of Pns11 and Lamp1 prevents the formation of autolysosomes, one would also need to perform an experiment to knockdown Lamp1 using dsLamp1. With a transient loss of Lamp1, one might expect to see an increase in RGDV and the number of autophagosomes in the midgut of viruliferous insects. In the absence of evidence supported by the additional experiments mentioned above, the model in Fig. 7 should be tempered by a measure of caution based on the preliminary nature of the key conclusion being presented to the reader.

6) Discussion L346-L348 presents an over-interpretation of the results. The authors have only shown immuno-EM data (micrographs) suggesting the movement of phagosomes between gut epithelial junctions, and EM micrographs of membranous vesicles fused with the APL of the salivary gland. Therefore, a more appropriate conclusion ought to be “…..to form membranous structures implicated to be involved in viral spread within the regions that we have examined”.

**Part III – Minor Issues: Editorial and Data Presentation Modifications**

Reviewer #1: - Page 3 line 77, it should be ATG8-I.

- Page 8, lines 237-239: Figure 5A shows the virus (Pns11) gene expression level. However, the sentence in the text does not mention this at all. It only talks about the ATG5 and ATG8. Either add the virus part or remove figure 5A in the parenthesis. The same for the following sentence. Figure 5A needs to be added in the parenthesis. As well

- Page 8, lines 235 and 247: What was the reason to use nymphs in two different life stages (2nd and 3rd) for microinjection?

- Page 11 lines 340-343: Add the “by an unknown mechanism” to the end of the sentence.

- Figure S2: ATG3 should be italic.

Reviewer #2: Fig. 1. What is the purpose of panel A? is it a control section derived from a healthy insect, or lower magnification of B? It would be helpful to add a little more details here, and label the ruler size. In B to F, instead of using arrows of different colors, why not simply label them (one each) as “gold particle”, “double membrane”, “virion”? Also add the antibody used on respective panels (e.g. C – RGDV P8 Ab).

Ln37, delete “in vivo” – redundant.

Ln38, add coma after “dwarf virus”

Ln39, change “within” to “of the”

Ln41-42, “viral nonstructural protein Pns11 induces autophagy and embeds itself in the autophagosome membrane.”

Ln43, delete “the development of”.

Ln47-48, “……to induce the formation of autophagosomes, the latter then enclose virions, and facilitate viral spread inside the insect bodies”.

Ln48-50, be specific – how does Psn11 causes Lamp1 down-regulation?

Ln54, delete “can”

Ln57, either “and” or “thus” has to go. Redundant.

Ln60, “mediating the down-regulation expression” – “down-regulating the expression” (same in Ln49).

Ln139, delete “characteristics”; add coma at the end of the line.

Ln140, “virus-infected” – “RGDV-infected”

Ln142, end, delete “were”

Ln143, add “an” before “RGDV-specific”.

Ln149, change “ATGs expression” to “the expression of ATGs” (to be more accurate, it should be “the mRNA levels of several ATGs”

Ln150, changed “increased” to “induced”; between “while” and “p62”, and “that of”

Ln151, delete “expression”; add coma after “stable”.

Ln152, “band” should be “protein level”.

Ln154, why “endogenous”? does the virus also encode an ATG8 homolog?

Ln166, “the component” should be “a component”

Reviewer #3: L170-L171: ‘immunoelectron microscopy was first employed to investigate which viral nonstructural proteins (Pns4,….or Pns 12) of RGDV localized to autophagosomes’. What were the results for Pns-4, -7, -9, -10, and -12)? Even if the results were negative, they should be mentioned.

L189-L191: “Taken together, RGDV can…….finally facilitating viral transmission by insect vectors (Fig 2F and 2G)’. These two figures are illustrative drawings of the presumed circulative pathway taken by the virus. To conclude that the autophagosomes facilitate viral transmission, one would have to link the RNAi knockdown studies of ATG5 to RGDV transmission study outcomes, but these studies are not reported in this manuscript.

L236: “with a mixture of purified viruses”. How were the viruses purified? This is not mentioned in the Materials and Methods.

MATERIALS AND METHODS

Sequence information of all primers used in the study should be provided, examples include those used for making the bacmid constructs (L429), the constructs used for GST pull-down assays (L452), and the constructs used for Y2H analysis (L416).

L387: How were total proteins extracted from the insects?

L432: Information on the construction of the recombinant bacmid constructs is needed.

L440: Information on the construction of GFP-ATG8 is needed.

L447-L450: There is a lot of overlapping materials and methods here and in L395 (“Examination of autophagy by immunofluorescence microscopy”) that can be consolidated.

L480: for all experiments that involved the use of statistical analyses, the number of replicates used in the studies need to be provided either in the figure legend or in the text.

FIGURE LEGENDS

L617-L618: Figure 1 legend. What are the green arrows pointing at?

L630: Figure 1 legend. How many times was the experiment involving the observation of 10 cells repeated?

L638-L640: this is a repetition of the previous sentence.

L644: Figure 2E. The panels in Fig. 2E are micrographs of the insect salivary glands without any immunogold labeling unlike in Fig. 2C (micrographs of the midgut invaded by RGDV). Without the immunolabeling, it is difficult to be convinced that those autophagosomes contained RGDV.

L650-L653: what does “VM” represent in Fig. 2C?

L679-L680: antibodies to these proteins were used i.e. anti-His-rhodamine, anti-Flag-rhodamine or anti-Pns11-rhodamine.

L697: Figure 5. What is the mock treatment? Based on the results, it looks like the mock treatment refers to non-viruliferous insects that were not subjected to any dsRNA treatment.

L710-L713: Figure 6. What do the white arrows indicate?

A SELECTION OF EDITORIAL ISSUES/MISTAKES

L274: “positive RNA viruses”

L283: “infection in the insect intestinal, disseminate…”

L345: “we demonstrates”

L370: “antibodies were then directly conjugated FITC”

L388 and L392: “The bands were then transferred..”. What bands? Please use the proper technical term to describe proteins separated by SDS-PAGE.

L463: “In vivo knocking down ATG5”

L464-L465: “..the functional roles of ATG5 or Pns11 in the autophagy inducing, the in vivo knocking down assay was conducted”

Figure 7: “Autolysome”

PLOS authors have the option to publish the peer review history of their article (what does this mean?). If published, this will include your full peer review and any attached files.

Reviewer #1: No

Reviewer #2: No

Reviewer #3: No
---

## [Editor Report · Decision Letter 1]

2 Mar 2022

Dear Dr Wei,

Thank you very much for submitting your manuscript "A nonstructural protein encoded by a rice reovirus induces an incomplete autophagy to promote viral spread in insect vectors" for consideration at PLOS Pathogens. As with all papers reviewed by the journal, your manuscript was reviewed by members of the editorial board and by several independent reviewers. The reviewers appreciated the attention to an important topic. Based on the reviews, we are likely to accept this manuscript for publication, providing that you modify the manuscript according to the review recommendations.

The authors have done a good job in responding to many of the suggestions of reviewers. However, in one area, addressed by both reviewers 2 and 3, they did not do an adequate job. The writing in several sections is still not carefully done. The authors need to have an expert correct the language mistakes throughout this manuscript, including the figure legends. For example just in the abstract, author summary and first paragraph of the introduction, here are the numerous minor language errors:

Line 50, membranes should be plural

51, remodel should be plural, insert the, it should say “the insect vector”.

Line 61, membrane should be plural

Line 76, ATG 8-1 which is…insert which as shown

Line 77, conjugate should be conjugated

Line 78, membranes should be plural

Line 81, membranes should be plural

Line 83, anautolysosomes should be autolysosomes

Line 84, protein should be plural

Line 85, should be “and protect”

Line 86, delete the word “was”, and consist should be plural

Line 87, insert “is”….and is

Line 89 is an improper sentence.

There are many more of these simple errors throughout the manuscript which must be corrected.

Other comments:

On line 209 - 215 the authors present data showing reduction of GFP-ATG8 when it is expressed by a baculovirus in Sf9 cells with pns11, they say increased accumulation of GFP-ATG8 and conversion to GFP-ATG8-II (Fig 3A). But for 3B they say increased expression, I think expression is the wrong word and accumulation/conversion are correct.

Somewhat similarly to above, on lines 247 – 261, they do additional experiments with dsRNA RNA-I against ATG5. I think the data are good, but I think they use some incorrect words. I think they see decreased ATG8 and ATG8-II accumulation, I don’t think they see an effect directly on conversion of ATG8 to ATG8-II. If there is less ATG8, there of course will be less accumulation of ATG8-II. Thus I would delete the word conversion.

For the data presented in lines 263 – 273, they argue that targeting pns11 by pns11 dsRNA-RNAi has the specific effects on levels of ATG5, ATG8-I/ATG8-II conversion, and the average number of ATG8-Pns11 puncta in midgut epithelial cells, and RGDV transmission. This is true, but it probably is not a result of specifically knocking down pns11, but by knocking down the pns11 RNA they are inhibiting RGDV replication and accumulation. They need to address this.

Line 308, all viruses are intracellular

Line 388, change conclude to speculate.

Sincerely,

Bryce W. Falk

Guest Editor

PLOS Pathogens

Peter Nagy

Section Editor

PLOS Pathogens

Kasturi Haldar

Editor-in-Chief

PLOS Pathogens

orcid.org/0000-0001-5065-158X

Michael Malim

Editor-in-Chief

PLOS Pathogens

orcid.org/0000-0002-7699-2064

The authors have done a good job in responding to many of the suggestions of reviewers. However, in one area, addressed by both reviewers 2 and 3, they did not do an adequate job. The writing in several sections is still not carefully done. The authors need to have an expert correct the language mistakes throughout this manuscript, including the figure legends. For example just in the abstract, author summary and first paragraph of the introduction, here are the numerous minor language errors:

Line 50, membranes should be plural

51, remodel should be plural, insert the, it should say “the insect vector”.

Line 61, membrane should be plural

Line 76, ATG 8-1 which is…insert which as shown

Line 77, conjugate should be conjugated

Line 78, membranes should be plural

Line 81, membranes should be plural

Line 83, anautolysosomes should be autolysosomes

Line 84, protein should be plural

Line 85, should be “and protect”

Line 86, delete the word “was”, and consist should be plural

Line 87, insert “is”….and is

Line 89 is an improper sentence.

There are many more of these simple errors throughout the manuscript which must be corrected.

Other comments:

On line 209 - 215 the authors present data showing reduction of GFP-ATG8 when it is expressed by a baculovirus in Sf9 cells with pns11, they say increased accumulation of GFP-ATG8 and conversion to GFP-ATG8-II (Fig 3A). But for 3B they say increased expression, I think expression is the wrong word and accumulation/conversion are correct.

Somewhat similarly to above, on lines 247 – 261, they do additional experiments with dsRNA RNA-I against ATG5. I think the data are good, but I think they use some incorrect words. I think they see decreased ATG8 and ATG8-II accumulation, I don’t think they see an effect directly on conversion of ATG8 to ATG8-II. If there is less ATG8, there of course will be less accumulation of ATG8-II. Thus I would delete the word conversion.

For the data presented in lines 263 – 273, they argue that targeting pns11 by pns11 dsRNA-RNAi has the specific effects on levels of ATG5, ATG8-I/ATG8-II conversion, and the average number of ATG8-Pns11 puncta in midgut epithelial cells, and RGDV transmission. This is true, but it probably is not a result of specifically knocking down pns11, but by knocking down the pns11 RNA they are inhibiting RGDV replication and accumulation. They need to address this.

Line 308, all viruses are intracellular

Line 388, change conclude to speculate.

Reviewer Comments (if any, and for reference):

Figure Files:

Data Requirements:

Reproducibility:

References:

---

## [Editor Report · Decision Letter 2]

6 Apr 2022

Dear Dr Wei,

We are pleased to inform you that your manuscript 'A nonstructural protein encoded by a rice reovirus induces an incomplete autophagy to promote viral spread in insect vectors' has been provisionally accepted for publication in PLOS Pathogens.

Best regards,

Bryce W. Falk

Guest Editor

PLOS Pathogens

Peter Nagy

Section Editor

PLOS Pathogens

Kasturi Haldar

Editor-in-Chief

PLOS Pathogens

orcid.org/0000-0001-5065-158X

Michael Malim

Editor-in-Chief

PLOS Pathogens

orcid.org/0000-0002-7699-2064

The authors have done a good job responding to suggestions and making corrections.
---

## [Editor Report · Acceptance letter]

2 May 2022

Dear Dr Wei,

We are delighted to inform you that your manuscript, "A nonstructural protein encoded by a rice reovirus induces an incomplete autophagy to promote viral spread in insect vectors," has been formally accepted for publication in PLOS Pathogens.

Best regards,

Kasturi Haldar

Editor-in-Chief

PLOS Pathogens

orcid.org/0000-0001-5065-158X

Michael Malim

Editor-in-Chief

PLOS Pathogens

orcid.org/0000-0002-7699-2064